# CONTEXTUAL IMAGE PARSING VIA PANOPTIC SEGMENT SORTING

## ABSTRACT

Visual context is versatile and hard to describe or label precisely. We aim to leverage the densely labeled task, image parsing, a.k.a panoptic segmentation, to learn a model that encodes and discovers object-centric context. Most existing approaches based on deep learning tackle image parsing via fusion of pixel-wise classification and instance masks from two sub-networks. Such approaches isolate things from stuff and fuse the semantic and instance masks in the later stage. To encode object-centric context inherently, we propose a metric learning framework, Panoptic Segment Sorting, that is directly trained with stuff and things jointly. Our key insight is to make the panoptic embeddings separate every instance so that the model automatically learns to leverage visual context as many instances across different images appear similar. We show that the context of our model's retrieved instances is more consistent relatively by $13.7\%$, further demonstrating its ability to discover novel context *unsupervisedly*. Our overall framework also achieves competitive performance across standard panoptic segmentation metrics amongst the state-of-the-art methods on two large datasets, Cityscapes and PASCAL VOC. These promising results suggest that pixel-wise embeddings can not only inject new understanding into panoptic segmentation but potentially serve for other tasks such as modeling instance relationships.

## 1 INTRODUCTION

Visual context is versatile and hard to describe or label precisely, yet it is critical for humans (Medin & Schaffer, 1978) to recognize objects quickly. More importantly, objects in different contexts carry different meanings. For example, pedestrians walking in crosswalks should receive more attention than on sidewalks to a driver. However, it is almost impossible to categorize objects with different contexts as the change can be subtle yet dramatic: A pedestrian is more likely in danger if walking in front of a car than by a car, where both a person and a car appear together. We are thus motivated to propose a model that automatically encodes and discovers object visual context by leveraging a densely labeled task, panoptic segmentation.

Panoptic segmentation (Kirillov et al., 2019b), a.k.a., image parsing (Tu et al., 2005), is to segment an image into its constituent visual patterns with both semantic and instance labels. The major challenge lies in delineating different instances while associating them with semantic categories. For example, one has to segment two side-by-side cars apart while still being able to classify them as the same category. Most existing approaches (Kirillov et al., 2019a; Xiong et al., 2019; Yang et al., 2019; Cheng et al., 2020; Li et al., 2020) tackle these two aspects via two sub-networks, instance and semantic segmentation branches. The advantage of such approaches is that each branch can cater to one aspect and achieves high performance. Additional modules for integrating things and stuff are needed to resolve the disagreements between two branches. Yet for object visual context, things and stuff are two integral parts. Hence, we aim to propose a framework that unifies these two seemingly competing aspects and thus encodes visual context inherently.

Our framework is inspired by the perceptual organization view (Biederman, 1987): Humans perceive a scene by breaking it down into visual groups and structures; repeated structures are then associated for cognitive recognition. Our key insight is to separate everything first and group visually similar components later. The grouping takes place within an image and across images. Within an image,

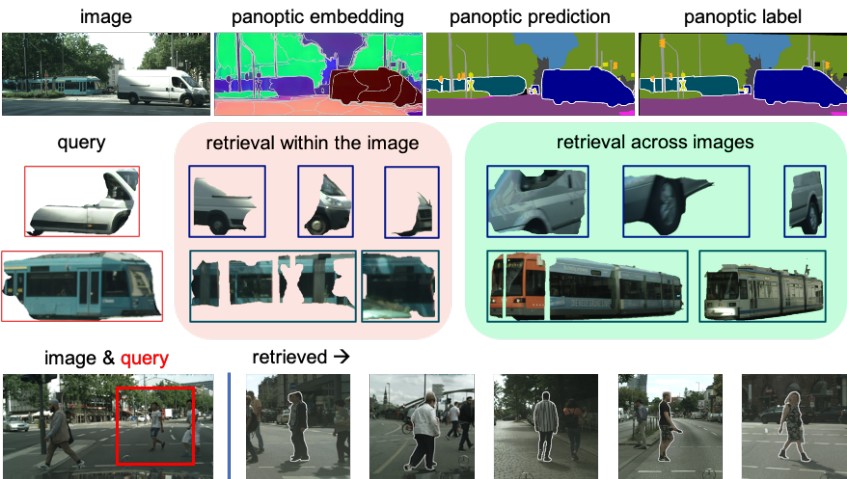

**Figure 1: Top row from left to right:** input image, panoptic embeddings, panoptic predictions, and panoptic labels. We overlay panoptic embeddings with the resultant over-segmentation boundaries. **Middle row:** After extracting panoptic embeddings from a CNN and the resultant over-segmentation, we use the segment prototype features to find nearest neighbors, within the image (middle) or across images (right), of each query segment (in red). These retrieval results probe what's learned in the embedding space. **Bottom row:** an example of contex specific instance retrieval results, where pedestrians crossing an intersection are discovered *unsupervisedly*.

visually similar segments are merged to form instances; across images, visually familiar segments are associated to create semantics, as illustrated in Fig. 1.

We carry out this idea by building an end-to-end trained pixel-wise embedding framework. Each pixel in an image is mapped via a CNN to a feature in latent space, and nearby features indicate pixels belonging to the same instance. This framework is therefore a non-parametric model at the segment and instance levels as its complexity scales with number of segments and instances, *i.e.,* exemplars. Particularly, by forcing all the instances to separate, the model has to utilize all the possible visual and semantic information. The model thus learns to separate instances by not only their appearances but also their surroundings, or visual context. A major difference between our model and others is the metric learning perspective: Our model trains with a contrastive loss that captures pixel-to-segment relationships while others trains with pixel-wise classification that predicts category or instance directly. As a result, the learned panoptic embeddings can discover instances under similar context, as in Figure 1 bottom row.

Specifically, we adapt the Segment Sorting approach (Hwang et al., 2019b) to panoptic segmentation by sorting segments according to both of its semantic and instance labels, hence dubbed Panoptic Segment Sorting (PSS). Such trained pixel-wise embeddings thus encode both semantic and instance information. We then predict each segment's semantic label by simply mapping and classifying its prototype feature with a softmax classifier. We also propose a corresponding clustering algorithm to merge segments into instances with a nearest neighbor criterion (Sarfraz et al., 2019). To alleviate the problem of instances with various scales, we further equip our framework with hybrid scale exemplars during training and dynamic partitioning during inference. Finally, we facilitate the merging process with a seeding branch that predicts the center of each instance.

As a result, we demonstrate that the contexts of instances retrieved by our panoptic embeddings are more consistent relatively by 13.7% while achieving competitive performance amongst the state-of-the-art on two datasets, Cityscapes (Cordts et al., 2016) and PASCAL VOC (Everingham et al.). These promising results suggest that Panoptic Segment Sorting or pixel-wise embeddings can not only inject new understanding into panoptic segmentation but potentially serve as a foundation for other tasks such as discovering novel contexts or modeling instance relationships.

## 2    RELATED WORK

**Image parsing and panoptic segmentation.** The task of image parsing is first introduced in Tu et al. (2005), where they formulate the solution in a Bayesian framework and construct a parsing

graph as output. Since then, a lot of work has attempted to solve holistic scene understanding (Zhu & Mumford (2007); Malisiewicz & Efros (2008); Tighe & Lazebnik (2013); Rabinovich et al. (2007); Yao et al. (2012)). Recently, Kirillov et al. (2019b) reintroduce image parsing in the context of deep learning with large-scale datasets and new evaluation metric, renaming the task as panoptic segmentation as to unify the well-developed semantic and instance segmentation. Many research efforts (Li et al., 2018b; Kirillov et al., 2019a; Xiong et al., 2019; Porzi et al., 2019; Yang et al., 2019; Liu et al., 2019; Li et al., 2019; 2018a; Gao et al., 2019; Chen et al., 2020; Wu et al., 2020; Wang et al., 2020; Li et al., 2020) have followed quickly. The common approaches embrace the concept of unifying instance and semantic segmentation by integrating the time-tested object proposal and segmentation framework popularized by Mask R-CNN (He et al., 2017).

**Instance segmentation.** This task is generally approached by two camps of solutions: top-down or bottom-up. The top-down approaches (Dai et al., 2016; Li et al., 2017; Dai et al., 2017; He et al., 2017; Chen et al., 2018a; Liu et al., 2018a) adopt a two-stage framework where the bounding boxes are proposed by a detection network (Ren et al., 2015) and the segmentation masks are produced by an add-on head. The bottom-up approaches (Carreira & Sminchisescu, 2011; Arbeláez et al., 2014; Pinheiro et al., 2015; 2016; Bai & Urtasun, 2017; Liu et al., 2017a; Kirillov et al., 2017; Newell et al., 2017; Fathi et al., 2017; Kendall et al., 2018; Liu et al., 2018b; Papandreou et al., 2018; Zhou et al., 2019) predict and encode pair-wise relationships in various forms and segment the instance accordingly.

**Instance context.** Instance contexts and relationships are explored mainly to enhance the detection performance. Earlier work (Malisiewicz & Efros, 2009) models the appearances and 2D spatial context as a graph. Recently, researchers integrate graphs (Chen et al., 2018c) or spatial memory (Chen & Gupta, 2017) into the deep learning framework. The distinction of our work is that our model does not explicitly model contexts yet is able to discovers novel contexts automatically.

**Semantic segmentation.** Current state-of-the-art semantic segmentation approaches develop from fully convolutional networks (Long et al., 2015; Chen et al., 2016), with various innovations. Incorporating contextual information (Ronneberger et al., 2015; Yu & Koltun, 2016; Xie et al., 2016; Zhao et al., 2017; Chen et al., 2017; 2018b), and encoding pair-wise relationships (Zheng et al., 2015; Bertasius et al., 2016; Liu et al., 2017b; Bertasius et al., 2017; Maire et al., 2016; Mostajabi et al., 2018; Kong & Fowlkes, 2018; Ke et al., 2018; Hwang et al., 2019a;b) are the two major research lines.

**Non-parametric segmentation.** Prior to deep learning's emergence, non-parametric models (Russell et al., 2009; Tighe & Lazebnik, 2010; Liu et al., 2011) usually use hand-craft features with statistical models or graphical models to segment images with pixel-wise labels. Deep metric learning methods (Fathi et al., 2017; Neven et al., 2019) for instance segmentation emphasize the simplicity and fast computation. More recently, inspired by non-parametric models (Wu et al., 2018b;a) for image recognition, SegSort (Hwang et al., 2019b), upon which our work is built, captures pixel-to-segment relationships via pixel-wise embeddings, proposing the first deep non-parametric semantic segmentation in both supervised and unsupervised settings.

## 3 METHOD

Our end-to-end framework consists of a major SegSort branch and a seeding branch, both of which share one backbone network that generates multi-scale pixel-wise features. The SegSort branch outputs pixel-wise panoptic embeddings, which encode both semantic and instance information and are thus used to discover instance-centric context. The over-segmentations induced by the embeddings are then merged into instances and segments are classified by a softmax classifier. The seeding branch predicts the center of instances, which guide the merging process to reduce false positives. The overall framework is illustrated in Figure 2.

This section is organized as follows. We first briefly review the Segment Sorting framework for semantic segmentation in Sec. 3.1. We then describe how to extend it for panoptic segmentation in Sec. 3.2. In Sec. 3.3, we further develop a dynamic partitioning mechanism to alleviate the problem of varying scales of instances. Finally, we briefly describe the seeding branch in Sec. 3.4 that helps decide the ownership of boundaries.

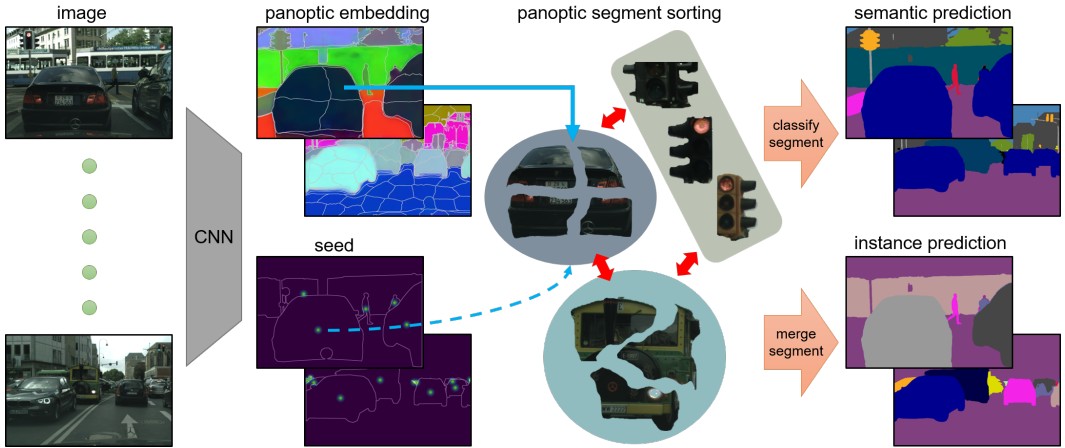

**Figure 2:** The overall end-to-end diagram of our proposed Panoptic Segment Sorting (PSS). We first over-segment an image with pixel-wise embeddings, extracted from a CNN. Each segment is represented by a prototype feature (average of pixel embeddings), which is then used for classifying segments (semantic predictions) and/or merging segments into an instance (instance predictions), whose features from the embeddings automatically encode object-centric context. An extra center seeding branch can faciliate the merging process by designating seed segments. The overall losses include (1) the SegSort loss (Hwang et al., 2019b) for embeddings, (2) the cross-entropy softmax loss for classification, and (3) the regression loss for seed locations.

## 3.1 SEGMENT SORTING

We briefly review the Segment Sorting (SegSort) approach proposed by Hwang et al. (2019b). Seg-Sort is an end-to-end optimization framework for non-parametric semantic segmentation. It produces pixel-wise semantic embeddings and their corresponding over-segmentation, each segment of which is then, during inference, assigned a semantic category via K-Nearest Neighbor search.

The basic idea of SegSort is assuming independent normal distributions (or von Mises-Fisher distributions for normalized embeddings) for individual segments, and seeking a maximum likelihood estimation of the feature mapping, so that the feature induced partitioning in the image and clustering across images provide maximum discrimination among segments. SegSort can be summarized as two components: spherical k-means clustering (Banerjee et al., 2005) and a maximum likelihood loss formulation with soft nighborhood assignments (Goldberger et al., 2005).

The spherical k-means clustering (Banerjee et al., 2005) alternates the expectation (E) and maximization (M) steps to partition the unit-length pixel-wise embeddings $\boldsymbol{v}$ of an image into $K$ regions $(\boldsymbol{R}_1, \ldots, \boldsymbol{R}_K)$. The M-step calculates the mean embedding direction of each region, or the *prototype* $\boldsymbol{\mu}_k = \frac{\sum_{i \in \boldsymbol{R}_k} \boldsymbol{v}_i}{||\sum_{i \in \boldsymbol{R}_k} \boldsymbol{v}_i||}$. The E-step assigns each pixel embedding $\boldsymbol{v}_i$ to a region $\boldsymbol{R}_k$ with nearest corresponding prototype $\boldsymbol{\mu}_k$, or $z_i = \arg\max_k \boldsymbol{\mu}_k^\top \boldsymbol{v}_i$, where $z_i$ is the segment index that the pixel $i$ is assigned. Note that the dot product on the right hand side is equivalent to cosine similarity as both $\boldsymbol{v}$ and $\boldsymbol{\mu}$ are of unit length. By alternating E- and M-steps, we over-segment an image.

After over-segmentation, one can derive a maximum likelihood loss with soft neighborhood assignments (Goldberger et al., 2005) to train the deep neural networks end-to-end. Interested readers are referred to the SegSort paper (Hwang et al., 2019b) for detailed derivation. The principle is to connect each pixel with one of its same-class segments, excluding its own segment, and to push away all the other segments in different classes. We define the corresponding probabilities given semantic segmentation ground truth labels as follows.

$$p(z_i = c^+ \mid \boldsymbol{v}_i, \Theta) = \frac{\exp(\kappa \boldsymbol{\mu}_{c^+}^\top \boldsymbol{v}_i)}{\sum_{l \neq c} \exp(\kappa \boldsymbol{\mu}_l^\top \boldsymbol{v}_i)}; \quad p(z_i = c \mid \boldsymbol{v}_i, \Theta) = 0, \tag{1}$$

where $\kappa$ is the concentration (around $\mu$) hyper-parameter in the von Mises-Fisher distributions, $c$ denotes the segment index to which the pixel $i$ is assigned, and $c^+$ denotes the segment index of any other same-class segment across all images in a batch. The final SegSort loss is therefore the

negative log-likelihood of a pixel $i$ selecting a same-class prototype as its neighbor:

$$L_{\text{SegSort}}^i = -\log \sum_{s \in C_i^+} p_\phi'(z_i = s \mid \boldsymbol{v}_i, \Theta) = -\log \frac{\sum_{s \in C_i^+} \exp(\kappa \boldsymbol{\mu}_s^\top \boldsymbol{v}_i)}{\sum_{l \neq c} \exp(\kappa \boldsymbol{\mu}_l^\top \boldsymbol{v}_i)}, \qquad (2)$$

where $C_i^+$ denotes the set of $c^+$ segment indices w.r.t. the pixel $i$, which is selected by the semantic segmentation ground truth labels. Minimizing this loss is equivalent to maximizing the expected number of pixels correctly classified by voting of their nearest neighbor prototypes.

## 3.2 PANOPTIC SEGMENT SORTING

Since the SegSort loss does not require a fixed number of classes as opposed to the conventional cross-entropy softmax loss, a way to extend it for instance discrimination is by changing the definition of ground truth labels and its corresponding selections of neighbor prototypes. In other words, we instead consider $c^+$ as the segment index of any other 'same-instance' segment. For stuff categories without instances, we consider all the segments in that class have the same instance label. With this modification, the SegSort loss in Eqn. 2 can be used to train panoptic embeddings.

Such trained embeddings, therefore, group each instance against all the other instances, regardless of their semantic categories. Still, since this loss pushes all the instances as far away as possible, visually similar instances are forced to stay closer on the hypersphere. We thus hypothesize two kinds of additional information are encoded: **(1)** The embeddings encode the semantic labels inherently as instances of the same class appear similar. To extract such information, we then stack two $1 \times 1$ convolutional layers on top of segment prototypes, followed by a softmax classifier to predict the semantic class of each segment. Note that no conflict between semantic and instance segmentations is introduced in this setting as they are built on the same over-segmentation. **(2)** The embeddings also encode object-centric context. This is because many instances (especially common objects like cars, persons, bikes, *etc.* ) across different images appear similar so the model has to leverage all possible information to push away all the instances of the same class.

Given the panoptic embeddings and the resultant over-segmentations, the challenge is to group segments into instances correctly during inference. We need two criteria: 1) how to merge segments, and 2) when to stop the merging. To align with the formulation of the SegSort loss, we adopt a nearest neighbor clustering criterion (Sarfraz et al., 2019) to greedily merge two segments $\boldsymbol{R}_m, \boldsymbol{R}_n$ with nearest prototypes, and stop the merging if the distance between two prototypes $\boldsymbol{\mu}_m, \boldsymbol{\mu}_n$ is greater than a threshold, or their dot product is less than a threshold $T_P$. The merging criteria can be summarized as:

$$\boldsymbol{R} = \{\boldsymbol{R}_m, \boldsymbol{R}_n\} \text{ if } \left(\mathcal{N}(\mu_m) = n \text{ or } \mathcal{N}(\mu_n) = m\right) \text{ and } \boldsymbol{\mu}_m^\top \boldsymbol{\mu}_n \geq T_P, \qquad (3)$$

where $\{\cdot, \cdot\}$ denotes merging segments, $\mathcal{N}(\cdot)$ denotes the index of the nearest neighbor prototype. We sort all the pairs of distances (dot products) of the prototypes in an image and consider merging greedily from the closest pair. We also update the new prototype after merging.

## 3.3 DYNAMIC PARTITIONING FOR HYBRID SCALE EXEMPLARS

The vanilla SegSort partitions an image into a fixed number of regions regardlessly. For semantic segmentation, this setting is reasonable as the number and sizes of homogeneous regions do not vary a lot from an image to another. However in instance segmentation, scales of objects can change drastically from 100 to 100K pixels. Oftentimes cluttered small instances will fall into one single segment, or even worse be included in another big instance. To alleviate this scale problem, we propose a hybrid scale setting for training and dynamic partitioning for inference accordingly. The illustrations can be found in Appendix.

During training, we consider regular embeddings $\boldsymbol{v}$ and their upscaled embeddings $\boldsymbol{v}^{(u)}$ by bilinear interpolation. The idea is to use the upscaled embeddings for small instances so that the gradient flows are finer. After the spherical k-mean clustering, we calculate segment prototypes using embeddings in different scales according to the instance sizes. Note that there is still only one prototype for each segment in the SegSort loss, be it either regular or upscaled.

During inference, the sizes of instances are unknown and have to be inferred. We notice that if a segment contains multiple small instances or multiple parts from a big instance, the corresponding

pixel embeddings are usually noisy, resulting in a low concentration. Therefore, we define an approximated concentration $\tilde{\kappa}_k$ of a segment $\boldsymbol{R}_k$ as $\tilde{\kappa}_k = \frac{||\sum_{i \in \boldsymbol{R}_k} \boldsymbol{v}_i||}{|\boldsymbol{R}_k|} \in [0, 1]$, where $|\boldsymbol{R}_k|$ denotes the number of pixels in the segment. If this value for a segment falls below a certain threshold $T_S$, we again partition this segment using the same spherical k-means (here $k = 4$ usually).

### 3.4 SEEDING BRANCH

We notice the boundaries between objects sometimes form their own segments, causing false positive instances. To remedy this issue, we build a second branch for predicting instance seeds, which are used for guiding the merging process, described in Sec. 3.2. We define seeds as the centers of instances and mark the segments that cover seeds as *seed segments*. For building this seeding branch, we follow closely the instance proposal branch in He et al. (2017); Xiong et al. (2019) and use the centers of the predicted bounding boxes as the seeds.

Once we predict the seeds and the corresponding seed segments, we perform a seeding variant of merging. The only modification of the merging of $\{\boldsymbol{R}_m, \boldsymbol{R}_n\}$ (in Eqn. 3) is that the segments to merge $\boldsymbol{R}_m \& \boldsymbol{R}_n$ are restricted to one seed segment and one non-seed segment; the merged segments are then marked as seed segments. Note that the merging only happens between same class segments. After this modification, all the boundary segments are then forced to be merged into one of the seed segments. The visualization of the merging processes can be found in the supplementary.

## 4 EXPERIMENTS

In this section, we demonstrate the efficacy of our framework through extensive experiments and analysis. We first describe the experimental setup in Sec. 4.1. We present the context specific instance retrieval results in Sec. 4.2. Finally in Sec. 4.3, we present the panoptic segmentation results. Hyper-parameters, ablation study, and more visual results such as panoptic predictions, context retrieval, and t-SNE (Maaten & Hinton, 2008) feature analysis, can be found in the Appendix.

### 4.1 EXPERIMENTAL SETUP

**Datasets.** We carry out experiments mainly on two datasets: Cityscapes and PASCAL VOC 2012.

Cityscapes (Cordts et al., 2016) is a dataset for semantic urban street scene understanding. $5,000$ high quality pixel-level finely annotated images are divided into training, validation, and testing sets with $2,975$ / $500$ / $1,525$ images, respectively. It defines 19 semantic categories containing flat, human, vehicle, construction, object, nature, *etc.* , of which 8 categories have instance labels.

PASCAL VOC 2012 (Everingham et al.) segmentation dataset contains 20 object categories and one background class. The augmented dataset contains $10,582$ (train) / $1,449$ (val) / $1,456$ (test) images. All the semantic classes, except for backgrounds, have instance labels.

**Network architecture.** We use the Feature Pyramid Networks (FPN) (Lin et al., 2017), with ResNet-50 (He et al., 2016) backbone pretrained on ImageNet, to provide the multi-scale pixel-wise features. For each of the seeding and panoptic embedding branch, we follow Xiong et al. (2019) by building three layers of deformable convolutional layers (Dai et al., 2017) (with shared weights across different scales) on top of each scale of FPN features. We then concatenate the multi-scale features, followed by a final fusion $1 \times 1$ convolutional layer. On top of the panoptic embeddings, we stack two $1 \times 1$ convolutional layers for the segment softmax classifier.

### 4.2 CONTEXT SPECIFIC INSTANCE RETRIEVAL

In this section, we experimentally verify our panoptic embeddings encode the object-centric context automatically.

**Discovery of novel context.** We retrieve the nearest neighbors of query instances on the Cityscapes validation set using their averaged embeddings. We notice that the retrieved instances are usually in similar context as the query. We showcase five interesting examples in the Appendix, *i.e.,* pedestrians crossing an intersecion (also in Figure 1) or walking next to cars, riders riding bikes together or next

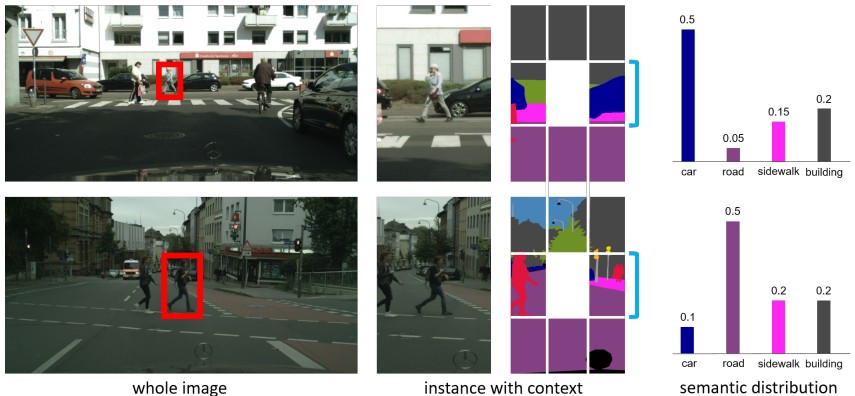

whole image          instance with context          semantic distribution

**Figure 3:** Illustration for evaluating the context similarity between two instances by comparing their semantic distributions in 8 extended regions. From left to right: whole image, instance of interest, semantic distribution of the middle right extended region. We calculate the symmetric KL-divergence between semantic distributions from corresponding regions as the context error.

to cars, and cluttered parked motorbikes. Note that these contexts are not given in the ground truth labels, yet our PSS can discover them unsupervisedly. We believe these examples are relevant in street scene understanding, especially for self-driving vehicles.

**Quantitative evaluation.** We wonder if such phenomena can be measured quantitatively. The challenge lies in the complicated scenarios and the lack of a complete label set. For example, crosswalks, which are labeled as roads, are visually similar as yet functionally different from roads. Furthermore, riding motorbikes next to cars is dangerous but difficult to describe precisely for annotating tasks.

We notice that the semantic category distribution of a larger patch captures some of such cases. For example, if there are multiple pedestrians nearby with cars around them, the chance of them walking on a crosswalk is higher. Such motivated, we propose to evaluate the context similarity between query and retrieval instances by comparing their semantic categories in 8 extended regions, as illustrated in Fig. 3.

To be specific, we denote the 8 neighbor regions (with the same size as the instance) as $B_j$ for $j = 1, \ldots, 8$. We calculate the semantic distribution in each region by the occupancy ratio of each class and denote it as $P_{B_j}$. That is, for each class, given a semantic label mask $S_P$, then $P_{B_j}^c = \frac{1}{|B_j|} \sum_{B_j [S_P = c]}$ for each category $c$, where $|B_j|$ denotes the area of region $B_j$. We then compare the semantic context distribution of the query $P_{B_j}^{(q)}$ against its $i$-th retrieval $P_{B_j}^{(r_i)}$ by calculating the symmetric KL divergence between the two, or

$$\text{CE} = \frac{1}{8K} \sum_{j=1}^{8} \sum_{i=1}^{K} \left( D_{\text{KL}}(P_{B_j}^{(q)}, P_{B_j}^{(r_i)}) + D_{\text{KL}}(P_{B_j}^{(r_i)}, P_{B_j}^{(q)}) \right), \tag{4}$$

where CE is our proposed metric, *Context Error*, and $K$ is the number of retrievals per query instance. If the reference probability is 0, the KL divergence will be invalid; in this case, we use a small probability $0.1$ instead. We set $K$ to 20 nearest neighbors.

We compute Context Error (CE) for each instance category, *i.e.,* , we restrict both query and retrieval to be a certain instance category. The final CE is the average errors of all instance categories. We compare our PSS against state-of-the-art UPSNet (Xiong et al., 2019) and summarize the results in Tab. 1. We observe PSS performs better in every category and reduces $13.7\%$ relative context errors.

| method | person | rider | car | truck | bus | train | mbike | bike | mean CE |
|---|---|---|---|---|---|---|---|---|---|
| UPSNet (Xiong et al., 2019) | 1.15 | 1.21 | 0.88 | 1.20 | 1.08 | 1.33 | 1.23 | 1.21 | 1.16 |
| **PSS** | 0.96 | 1.01 | 0.65 | 1.12 | 1.04 | 1.27 | 1.11 | 1.05 | 1.02 (**-13.7%**) |

**Table 1:** *Context Errors* (CE) on the Cityscapes (Cordts et al., 2016) validation set. We observe PSS performs better in every category and reduces $13.7\%$ relative context errors.

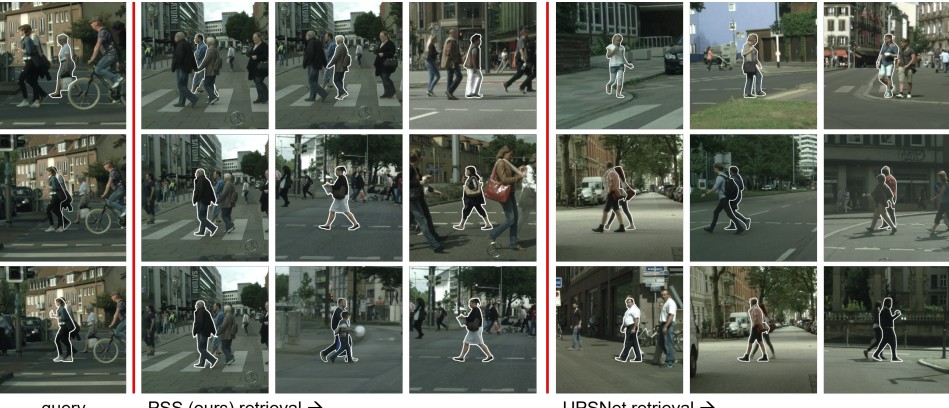

query    PSS (ours) retrieval →      UPSNet retrieval →

**Figure 4:** Visual comparison for context specific instance retrieval. We show 3 query examples (left) and their top retrieval results by our PSS (middle) and UPSNet (Xiong et al., 2019) (right), respectively. We observe that retrieved instances by PSS are usually in similar context or sometimes even from the same training image.

| method | backbone | PQ | PQ$^{Th}$ | PQ$^{St}$ |
|---|---|---|---|---|
| Li et al. (2018b) | ResNet-101 | 47.3 | 39.6 | 52.9 |
| DeeperLab (Yang et al., 2019) | Xception-71 | 56.5 | - | - |
| AUNet (Li et al., 2019) | ResNet-50 | 56.4 | 52.7 | 59.0 |
| SSAP (Gao et al., 2019) | ResNet-50 | 56.6 | 49.2 | - |
| Panoptic FPN (Kirillov et al., 2019a) | ResNet-50 | 57.7 | 51.6 | 62.2 |
| UPSNet (Xiong et al., 2019) | ResNet-50 | 59.3 | 54.6 | 62.7 |
| UPSNet* (Xiong et al., 2019) | ResNet-50 | 59.1 | 54.2 | 62.6 |
| **PSS** | ResNet-50 | 58.7 | 51.7 | **63.7** |

**Table 2:** Experimental results on the Cityscapes validation set. Our proposed framework PSS achieves competitive performance in PQ and outperforms all the other methods in PQ$^{St}$. * denotes retraining the model using released code; other results are copied from the published papers and '-' denotes missing metrics.

**Visual Comparison.** Next, we present the visual comparison in Fig 4 between our PSS and UPSNet using three query instances from the same validation image and display 3 retrieved instances for each network in the training set. We observe that our retrieved instances are usually in a similar context and are sometimes even from the same training image. It indicates that PSS encodes not only the appearances of an instance but also its nearby environment.

**Visual Context Cluster Analysis** We conduct visual context cluster analysis and visualize the results in Fig. 5. We first collect all the pedestrian prototypes in the Cityscapes training set. We plot their surrounding ground truth mask at their t-SNE feature locations and the aggregated density map. We observe interesting clusters such as pedestrians next to a car (center) and pedestrians alone on sidewalks (top left). We also notice some rare contexts on the middle left by examining the density map: a pedestrian is behind a clutter of a motorbike and a bike, which could lead to collision.

### 4.3 Panoptic Segmentation

**Main results on Cityscapes.** We summarize the main results on the Cityscapes validation set and compare with the state-of-the-art in Table 2. Our PSS achieves competitive performance in PQ (Panoptic Quality, explained in Appendix) and outperforms all the other methods in PQ$^{St}$. Notably, our framework performs particularly well in semantic segmentation related benchmarks.

**Main results on PASCAL VOC.** We summarize the main results on the PASCAL VOC validation set and compare with the state-of-the-art in Table 3. We show that PSS outperforms Li et al. (2018b) by 2% PQ even with a weaker backbone (ResNet-50 vs 101).

| method | backbone | PQ |
|---|---|---|
| Li et al. (2018b) | ResNet-101 | 62.7 |
| **PSS** | ResNet-50 | 64.8 |

**Table 3:** Experimental results on Pascal VOC 2012 validation set.

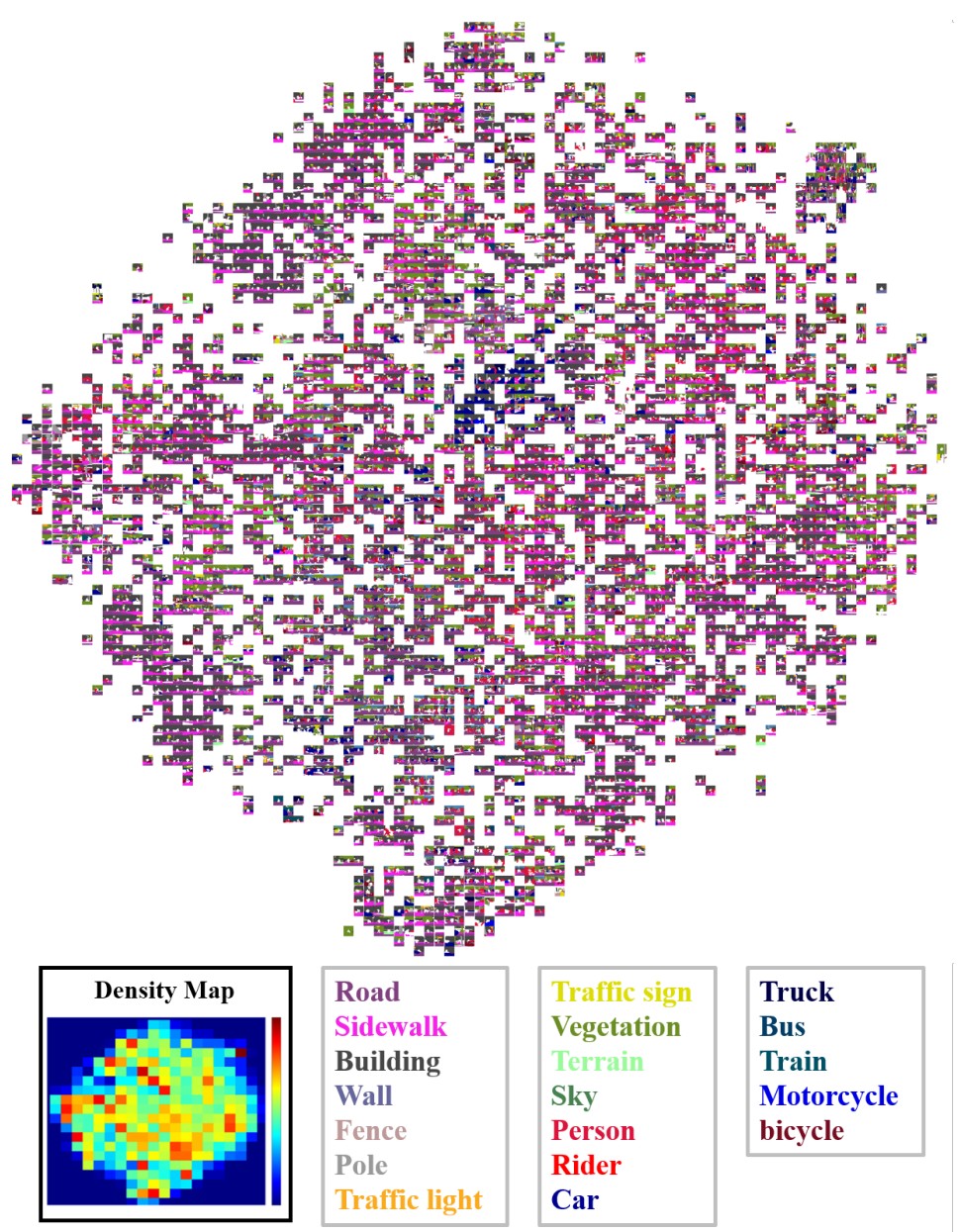

| Density Map | | | |
|---|---|---|---|
| | **Road** | **Traffic sign** | **Truck** |
| | **Sidewalk** | **Vegetation** | **Bus** |
| | **Building** | **Terrain** | **Train** |
| | **Wall** | **Sky** | **Motorcycle** |
| | **Fence** | **Person** | **bicycle** |
| | **Pole** | **Rider** | |
| | **Traffic light** | **Car** | |

**Figure 5:** Pedestrian-centric visual context cluster visualization (best viewed with zoom-in). We first collect all the pedestrian prototypes in the Cityscapes training set. We plot their surrounding ground truth mask at their t-SNE feature locations and the aggregated density map (bottom left). We observe interesting clusters such as pedestrians next to a car (center) and pedestrians alone on sidewalks (top left). We also notice some rare contexts on the middle left by examining the density map: a pedestrian is behind a clutter of a motorbike and a bike, which could possibly lead to collision.

## 5 SUMMARY

We presented the Panoptic Segment Sorting (PSS) framework for contextual image parsing. We aimed to encode and discover object-centric context automatically by unifying semantic and instance segmentation in the pixel-wise panoptic embeddings. We experimentally demonstrated such trained embeddings automatically encode object-centric context for better instance discrimination. PSS also achieved competitive performance amongst the state-of-the-art when equipped with hybrid scale exemplars, dynamic partitioning, and instance seeding. One future direction is to study how to explicitly disentangle the representations for instance appearances and contexts.

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

# A APPENDIX

We provide a metric learning approach to contextual image parsing. It is not only a unified approach to semantic segmentation and instance segmentation, but most excitingly also the first segmentation method that produces a learned feature representation directly applicable for more powerful context-specific image retrieval. Here we include more details on the following aspects:

- We present the visual results of novel context discovery in Section A.1.

- We present more interesting visual context comparisons in Section A.2.

- We present the visual results of our panoptic segmentation predictions on Cityscapes validation set in Section A.5.

- We visualize hybrid scale exemplars and merging processes with and without seeds, for further insights into our algorithm in Section A.3 and A.4, respectively.

- We analyze the car instance embeddings using t-SNE visualization in Section A.6.

- We provide detailed description of our experimental setup regarding hyper-parameters and ablation study in Section A.7.

## A.1 VISUAL RESULTS OF NOVEL CONTEXT DISCOVERY

As discussed in Sec. 4.2, we retrieve the nearest neighbors of query instances on the Cityscapes validation set using their averaged embeddings. Here, we showcase five interesting examples in Fig. 6, *i.e.,* pedestrians crossing an intersecion or walking next to cars, riders riding bikes together or next to cars, and cluttered parked motorbikes. Note that these contexts are not given in the ground truth labels, yet our PSS can discover them unsupervisedly. We believe these examples are relevant in street scene understanding, especially for self-driving vehicles.

## A.2 ADDITIONAL VISUAL COMPARISONS FOR CONTEXT SPECIFIC INSTANCE RETRIEVAL

We present more interesting visual comparisons for context specific instance retrieval in Fig. 7. It can be observed that PSS captures more context when retrieving instances. For examples, riding bikes along a sidewalk to its left, parked cars with a bike passing by, driving cars with pedestrians nearby, riding motorbikes behind a car, *etc.*

## A.3 ILLUSTRATION OF HYBRID SCALE EXEMPLARS

In Section 3.3 in the main paper, we describe the training process with hybrid scale exemplars. In short, when we train the panoptic embeddings with the SegSort loss, we select segment prototypes from embeddings of different scales, decided by their corresponding instance sizes. Particularly, we use upscaled embeddings for prototypes of small instances. We illustrate this process in Fig. 8.

## A.4 VISUALIZATION OF MERGING PROCESS

We describe the merging process (without and with seeds) in Section 3.2 and 3.4, respectively. To facilitate the understanding, we visualize a few intermediate steps in Fig. 9. We observe that once an instance composes of a single segment, its cosine similarity to its nearest prototype drops significantly. It indicates this merging process with threshold works well with our training objective. Also note that the merging usually grows from interior pixels towards boundaries.

## A.5 PANOPTIC PREDICTIONS

We present the visual results of our panoptic segmentation predictions on the Cityscapes validation set in Fig. 10. We observe that our semantic predictions capture well thin objects such as poles and traffic signs, even if many of them are labeled ignore. Also, the far away cars (in example 3) and pedestrians (in example 1) are also detected.

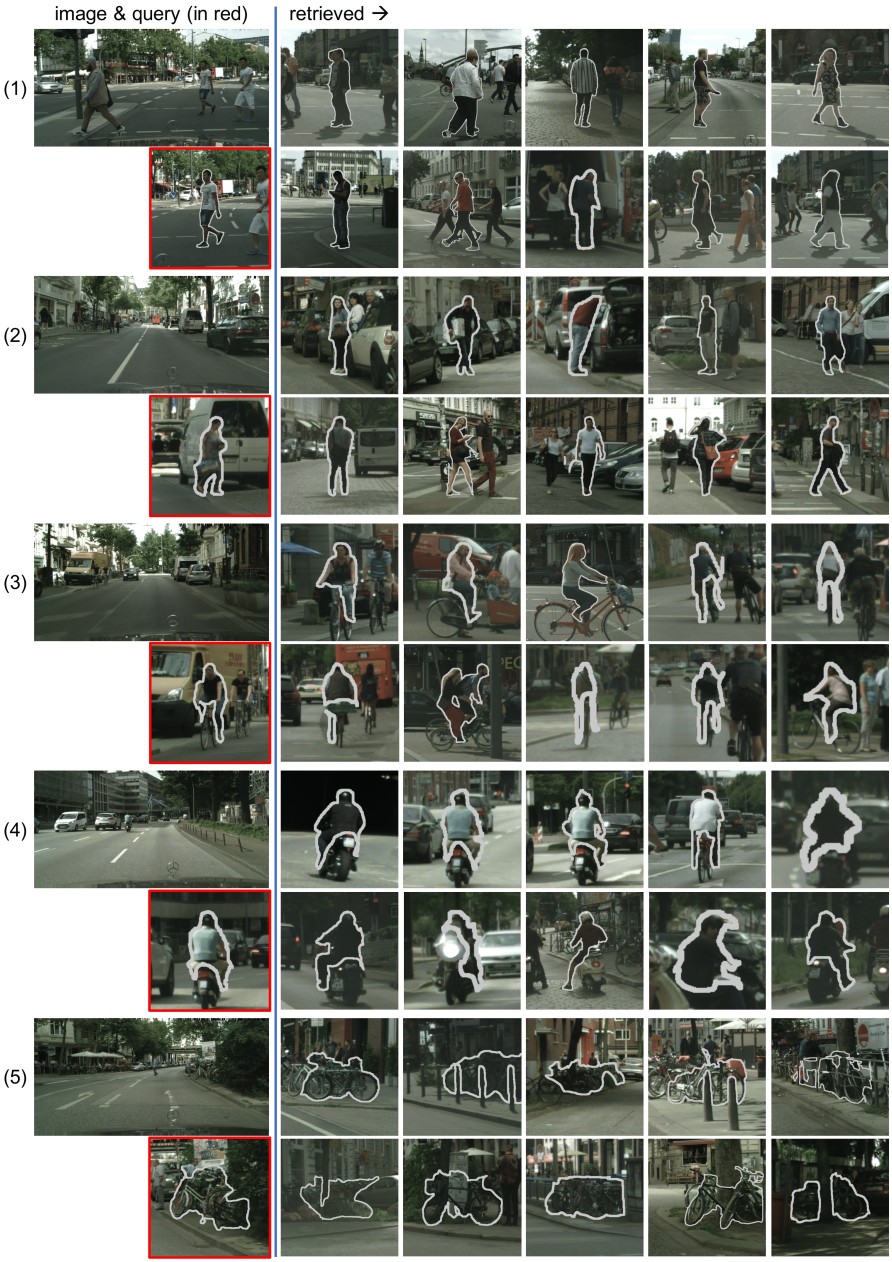

**Figure 6:** Retrieved the K-Nearest Neighbors for query instances. The image and query are on the left, and to their right are the retrieved instances (in the order of left to right, top to bottom). We demonstrate one kind of contexts in each example: 1) Pedestrians crossing an intersection. 2) Pedestrians walking next to cars. 3) Riders riding bikes together. 4) Riders riding motorbikes next to cars. 5) Parked motorbikes. Note that these contexts are not given in the ground truth labels, yet our PSS can discover them automatically. We believe these examples are relevant in street scene understanding, especially for self-driving vehicles.

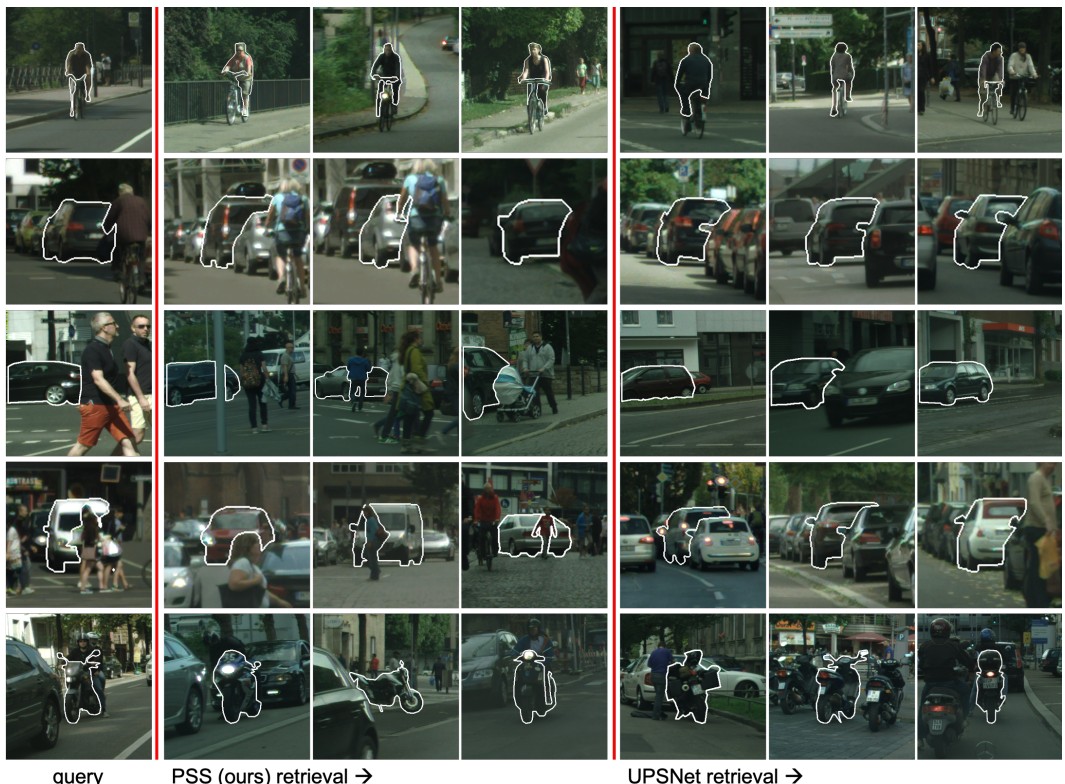

query          PSS (ours) retrieval →                    UPSNet retrieval →

**Figure 7:** More visual comparisons for context specific instance retrieval between PSS (ours) and UPSNet. It can be observed that PSS captures more context when retrieving instances. For examples, riding bikes along a sidewalk to its left, parked cars with a bike passing by, driving cars with pedestrians nearby, riding motorbikes behind a car, *etc.*

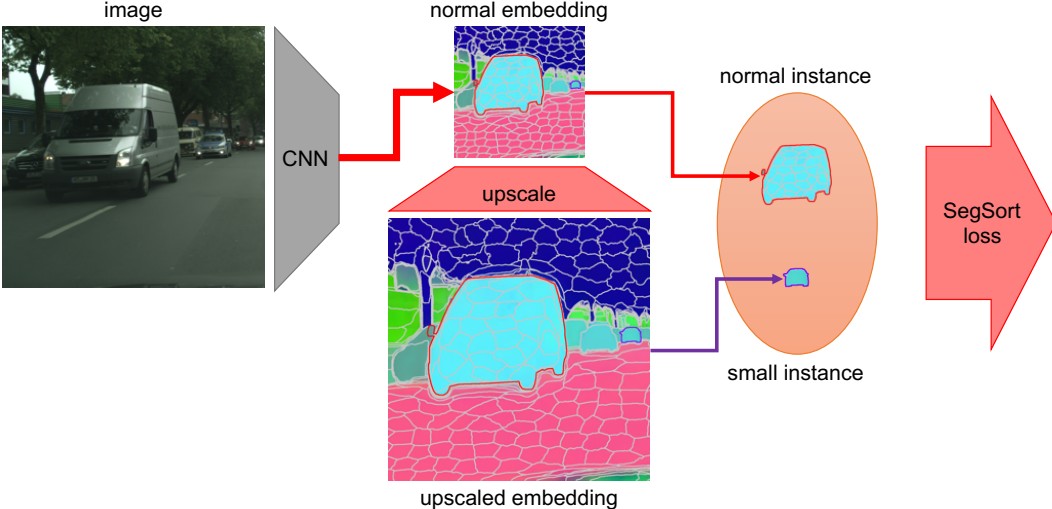

**Figure 8:** Illustration of SegSort training with hybrid scale exemplars in Section 3.3 in the main paper. The embeddings are overlayed with the predicted over-segmentation (with gray boundaries). We emphasize here one normal sized instance (outlined in red) and one small instance (outlined in purple), the prototypes of segments of each are collected from corresponding embeddings.

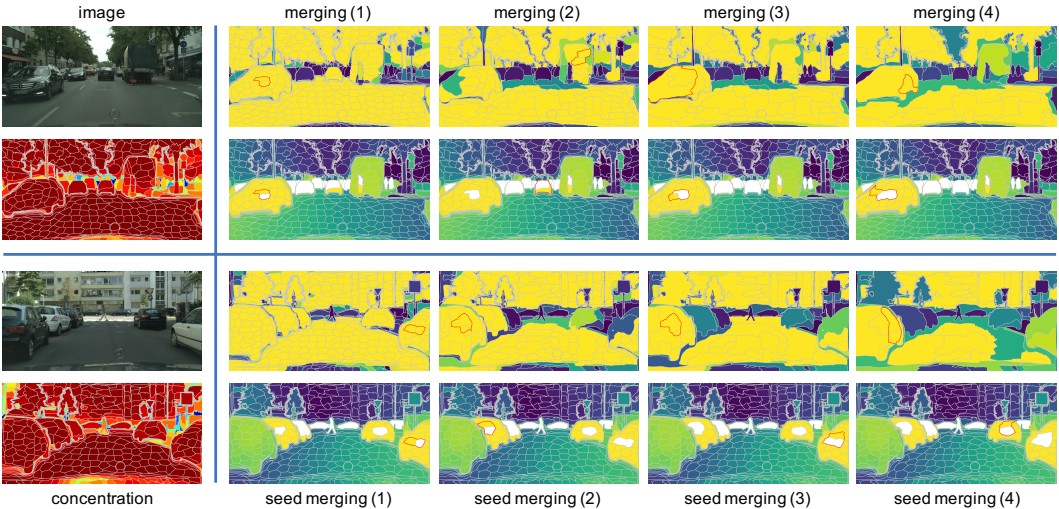

**Figure 9:** Visualization of the merging processes, without and with seeds, described in Section 3.2 and 3.4 in the main paper, respectively. On the left are the input images and approximated concentration per segment (overlayed with oversegmentation boundaries). To their right are merging processes without seeds (first row) and with seeds (second row). The red contours outline two segments to merge in this step. The viridis heat maps indicate the cosine similarity between each segment and its nearest prototype. (The white segments are seed segments.) The merging starts from the pair with the highest cosine similarity. We visualize a few steps in the beginning. Note that once an instance composes of a single segment, its cosine similarity to its nearest prototype drops significantly. It indicates this merging process with threshold works well with our training objective.

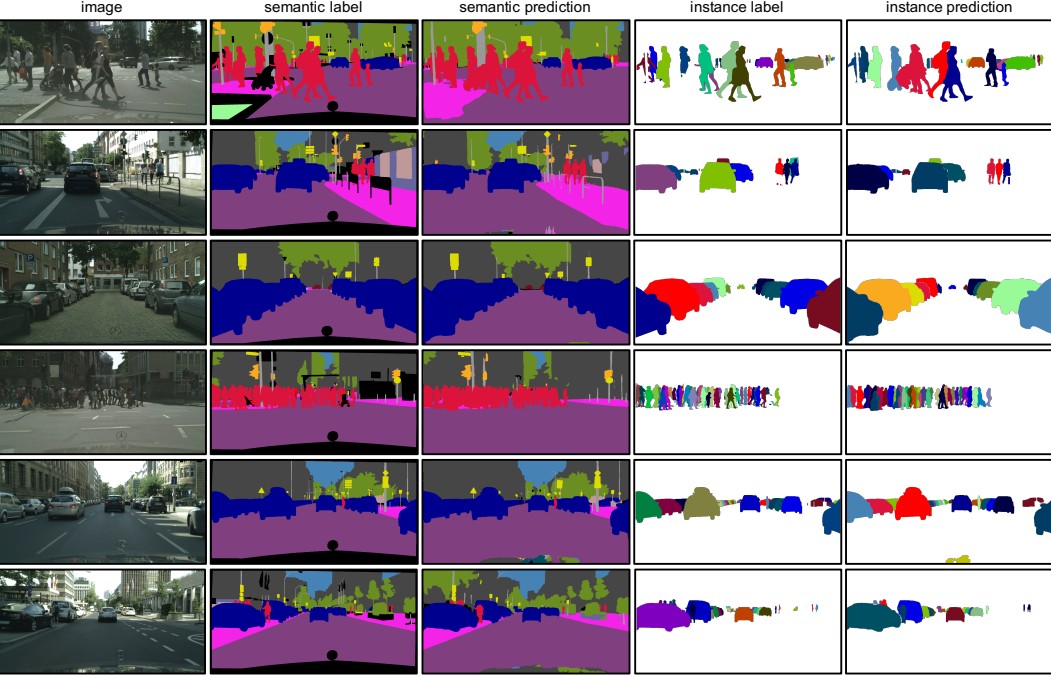

**Figure 10:** The panoptic segmentation visual results on the Cityscapes validation set of our proposed framework, Panoptic Segment Sorting. We observe that our semantic predictions capture well thin objects such as poles and traffic signs, even if many of them are labeled ignore. Also, the far away cars (in example 3) and pedestrians (in example 1) are also detected.

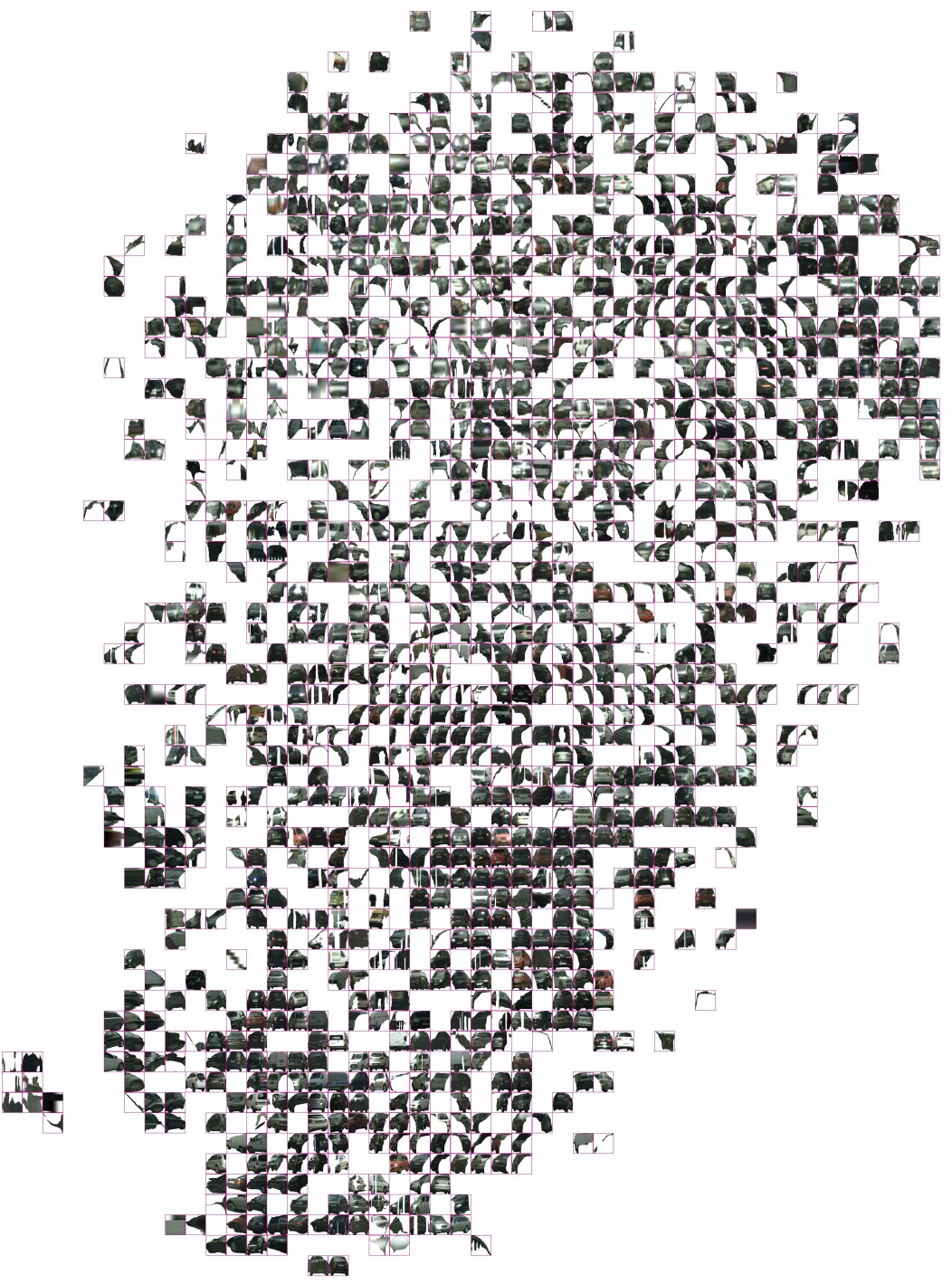

**Figure 11:** We visualize the prototype features of cars on the Cityscapes training set using tSNE. We observe that the features cluster certain parts or types of cars automatically. For example, front wheels appear on the bottom left corner and red cars in the center. Best viewed with zoom-in.

## A.6   T-SNE VISUALIZATION

We visualize the prototype features of cars on the Cityscapes training set using t-SNE in Fig. 11. We observe that the features cluster certain parts or types of cars automatically. For example, front wheels appear on the bottom left corner and red cars in the center.

| hybrid | dynamic | PQ | $PQ^S$ | $PQ^M$ | $PQ^L$ | mIoU |
|--------|---------|-----|--------|--------|--------|-------|
|        |         | 51.5 | 5.5 | 32.4 | 67.4 | 75.60 |
|        | ✓       | 52.4 | 7.3 | 31.5 | 67.1 | 76.16 |
| ✓      |         | 52.2 | 8.4 | 33.8 | 67.2 | 75.16 |
| ✓      | ✓       | 53.7 | 11.2 | 34.0 | 66.6 | 76.02 |

**Table 4:** Ablation study on hybrid scale exemplars and dynamic partition. 'Hybrid' denotes enabling hybrid scale exemplars during training and 'dynamic' denotes enabling dynamic partition during inference. The most significant gain is observed for small instances where $PQ^S$ increases by 5.7%. This verifies our claim for alleviating the instance scale problem.

| seeding | classifier | PQ | $PQ^S$ | $PQ^M$ | $PQ^L$ | mIoU |
|---------|-----------|-----|--------|--------|--------|-------|
|         |           | 44.0 | 9.3 | 25.4 | 55.6 | 61.66 |
| ✓       |           | 45.5 | 8.2 | 27.0 | 56.6 | 61.62 |
|         | ✓         | 50.9 | 12.9 | 33.4 | 63.6 | 75.52 |
| ✓       | ✓         | 53.7 | 11.2 | 34.0 | 66.6 | 76.02 |

**Table 5:** Ablation study on seeds and segment softmax classifier. Two convolutional layers (and a softmax classifier) to map the features onto semantic latent space are still necessary to produce good predictions (+7% in PQ). We also observe the seed predictions help the merging process and boost the performance (+2% in PQ).

## A.7 Hyper-parameters and Ablation Study

**Hyper-parameters.** We train the models on Cityscapes for 90k iterations. The batch size is set to 8, and the crop size to 704. On VOC dataset, the training iteration is 60k, the batch size is 12 and crop size is 512. We adopt the standard poly learning rate policy with the base learning rate, momentum, and weight decay as 0.002, 0.9, and 0.0001, respectively.

For Panoptic Segment Sort, we set the hyper-parameters for all the experiments as follow. The dimension of panoptic embedding is 64. The concentration is set as 8, and the iterations in K-Means is 10. On Cityscapes, we use 196 clusters in K-Means for training. During inference, we set the number of clusters as 500 and the threshold $T_S$ of dynamic partition as 0.99. On VOC dataset, the number of cluster is 64. We perform dynamic partition with $T_S$ as 0.99 for inference. The small instance area upper bound $A_p$ is set to 2048 and 512 for Cityscapes and VOC dataset.

For the seeding branch, we generate the Gaussian heatmap by setting the radius to 16 pixels. We train the pixels within the radius to predict the offsets. The overall loss weights for the SegSort loss, cross-entropy softmax loss, L1 and L2 seed location loss are 1, 1, 0.1 and 1.

**Post-processing.** We perform minimal post-processing on the final predictions. For stuff categories, we set the region with areas less than 2,048 to void. For thing categories, if non-seed segments after merging has area less than 4096, we set them to void as well.

**Ablation study on hybrid scale exemplars and dynamic partition.** The ablation study is summarized in Table 4. We enable and/or disable the hybrid scale exemplars during training and dynamic partition during inference. We train the network with only 30k iterations on Cityscapes for faster experimentation. 'Hybrid' denotes enabling hybrid scale exemplars and 'dynamic' denotes enabling dynamic partition. Each component will boost the performance by $0.7 - 0.9\%$ PQ and by $2.2\%$ PQ if both are enabled. The most significant gain is observed for small instances where $PQ^S$ increases by $5.7\%$. This ablation study verifies our claim for alleviating the instance scale problem.

**Ablation study on seeds and segment softmax classifier.** The ablation study is summarized in Table 5. We train the network with only 30k iterations on Cityscapes. We enable and/or disable the seeding branch and segment softmax classifier with two convolutional layers.

When the classifier is disabled, we perform K-Nearest Neighbor Search using the prototype features from the panoptic embeddings directly. It shows that two convolutional layers (and a softmax classifier) to map the features onto semantic latent space are still necessary to produce good classification predictions ($+7 \sim 8\%$ in PQ). Also, We observe the seed predictions help the merging process and boost the performance by $1.5 \sim 2.8\%$ in PQ.

