# OpenReview forum: "Contextual Image Parsing via Panoptic Segment Sorting"
_ICLR.cc/2021/Conference — Reject_

### Official Review · AnonReviewer3 · 2020-10-27

**Rating:** 6
**Confidence:** 3

**Review:**

##########################################################################

Summary:

This paper adapts Segment Sorting to panoptic segmentation and proposes a Panoptic Segment Sorting (PSS). The proposed method learns to sort segments according to both of its semantic and instance labels. The semantic label is acquired by simply mapping and classifying prototype feature and instances are formed by a clustering algorithm. A seeding branch is further used to guide merging and avoid false positives.

##########################################################################

Pros:

- The overall idea is very interesting, instead of using separate models (or branches) to learn instance segmentation and semantic segmentation, the proposed method directly learns a unified embedding that encodes both instance and semantic information. This make the model more consistent.
- It is also shown in the experiment that the panoptic embedding is able to encode object-centric context.

##########################################################################

Cons:

- It is still not clear to me how does the learned panoptic embedding able to encode object-centric context. Throughout the paper I don't see any specific design to optimize to learn context and the authors also do not provide theoretical explanation. If this is the case, I don't think it is suitable to name it "contextual image parsing" and put "discovering object-centric context" in the motivation (or the goal), since I think it is not well studied.
- If the proposed method indeed learns object-centric context, I think it is more reasonable to apply the method to tasks where object-centric context is important, e.g. relation detection.

##########################################################################

Typos:

- Last sentence is not finished in paragraph 2 of Section 1, "A major"

---

> ### Author Response · Authors · 2020-11-24
> **Thanks much for acknowledging the novelty of our idea and the experiments on object-centric contexts**
>
> Thanks much for acknowledging the novelty of our idea and the experiments on object-centric contexts!  Now we answer your questions.
>
> Q1: How does the embedding encode contexts?
> * The embedding of our model learns to encode contexts through contrastive learning.  As described in the Method section, we ask the model to separate all the instances in the training set.  However, many instances (especially common objects like cars, persons, bikes, etc.) across different images appear quite similar so the model has to leverage all possible information (i.e., visual contexts) to push apart all the instances of the same class.  In other words, we are mining co-occurrence statistics in fragments via contrastive training.  Such as attraction and repulsion relationships implicitly, rather than explicitly, encode the visual context, which is rather remarkable in terms of how the context is operationalized in a pure data-driven manner.
>
> Q2: Relation detection?
> * Great suggestion for the relation detection task!   While our method could be applicable for relation detection, it often captures versatile visual contexts that are hard to describe or label precisely, which we believe is currently beyond the relation detection with a fixed set of relationship labels.   That is, while in theory our method is capable of encoding relations, in practice benchmarking the strength is limited by such annotated datasets that can best demonstrate the merit of our approach.

---

### Official Review · AnonReviewer1 · 2020-10-28
**Interesting extension of SegSort, with unclear motivation**

**Rating:** 6
**Confidence:** 3

**Review:**

**Pros**

* Competitive results on benchmarks. The PQ results are good, with relatively few additional "tricks" (assuming they're not simply left out of experimental section).

* Complete ablation studies. This includes swapping in/out some proposed improvements to the underlying SegSort training, plus different components of the method used to construct panoptic segmentations from the oversegmentation.

* Makes additional interesting observations about the behavior of the model/properties of the embeddings, in looking at the "context."

**Cons**

* The motivation is somewhat unclear. The core "segment sorting" method is an unsupervised method. The method here only builds on the supervised variant of SegSort and introduces additional components that require supervision. So the main difference from existing panoptic segmentation methods is the use of an intermediate segmentation. The authors propose the embeddings are useful in "context specific instance retrieval," but the importance of this task is unproven. The fact that it seems to improve panoptic segmentation results is more concrete, but it is less clear *why* this method would yield this improvement.

* Description of hybrid scale exemplars is unclear, with very few details.

**Typos**

  * "objects in different context" -> "objects in different contexts" on page 1
  * Typo "differnet" on p1
  * Incomplete sentence "A major..." on p1
  * "equivalent to maximize the expected number" -> "equivalent to maximizing the expected number" on p5
  * Double periods after "segment" at top of p6
  * "to be certain [a] instance category" on p7

---

> ### Author Response · Authors · 2020-11-24
> **Thanks much for acknowledging our competitive results with complete ablation studies and recognizing the novelty about contextual image parsing**
>
> Thanks much for acknowledging our competitive results with complete ablation studies and recognizing the novelty about contextual image parsing!  We appreciate your catch on a few typos, which are now all fixed in the revised version.  Below are our answers to your doubts.
>
> Q1: Motivation? Importance of context retrieval? Why does it improve panoptic segmentation?
> * Our goal is not to simply beat the benchmarks on panoptic segmentation, but to use it to enable visual context encoding implicitly.   That is, to encode visual contexts, we need to understand objects and stuff; while developing our novel panoptic segmentation by studying the contrastive relationships between pixels and segments in terms of objects and stuff, we automatically capture the visual context implicitly.  While more work is needed to further explore visual contexts, to the best of our knowledge, there has not been any other work in panoptic segmentation that even remotely studies visual contexts.
>
> * There are at least 4 possible challenging and exciting applications of object-centric contexts.
> 1) Fine-grained semantic categories by object-centric contexts.  As suggested in the paper, pedestrians can be further categorized by where they appear (e.g. in a busy intersection vs. an empty sidewalk) or what actions they are taking (e.g. window shopping vs. hailing a taxi).
> 2) Data-driven scene context organization.  Scene contexts can be hard to describe in words and yet very useful in practical applications such as autonomous driving.  Our work can be used to actively slice the training data according to scenes of interest without any scene labels.  For example, tail events and corner cases can be identified from a large unlabeled dataset and new models can be trained to improve the accuracy of predictions for those rare yet critical cases.
> 3) Contexts for object recognition.  It has been long recognized that an object can be readily identified not by what foreground it is itself, but by what background it is embedded into (Torralba, IJCV 2003).  Object-centric contexts can be useful for speeding up and improving the accuracy of object detection and recognition.
> 4) Anomaly detection.  We can analyze the statistics of object-centric contexts for a specific query object category and find rare or abnormal events.  Please refer to our answer to R4’s Q3, where we add a new t-SNE visual analysis on page 9 in the main paper to showcase how our learned embedding copes with anomaly contexts.
>
> * The most relevant comparison is against another metric learning framework, ASPP, which we improve by 2% in terms of PQ on Cityscapes.  The major difference between ours and ASPP is that ASPP only uses metric learning (embedding) for the instance branch and still models semantic segmentation using a pixel-wise classification branch.  Ours, on the other hand, models both instance and semantic segmentation through a shared embedding, which makes the model learn more complete features for panoptic segmentation as a whole and enables better performance.
>
>
> Q2: Details of hybrid scale exemplars?
> * We illustrate the hybrid scale exemplars in Figure 7 in the supplementary.  During training, we consider regular embeddings and their upscaled (2x2) embeddings by bilinear interpolation. The idea is to use the upscaled embeddings for small instances so that the gradient flows are finer. We define a small instance as having less than 2048 pixels on Cityscapes.   After the spherical k-mean clustering, we calculate segment prototypes using embeddings at different scales (either regular or upscaled) according to the instance size.  The rest of the training remains unchanged.  We will provide a clearer description in the final version.

---

### Official Review · AnonReviewer4 · 2020-10-29
**Interesting idea but seems to lack focus**

**Rating:** 5
**Confidence:** 4

**Review:**

The paper presents a pixel-wise embedding strategy for panoptic segmentation, which aims to learn a pixel representation that encodes both semantic and instance information. To this end, the proposed method builds on top of the Segment Sorting approach and extends its contrastive loss to the instance level by utilizing panoptic supervision. To predict instance segmentation,  the paper also designs a merging process to cluster the pixels into instances, which further employs an object center prediction module for localization and a dynamic partition strategy to cope with scale variation.  This method is evaluated on two panotpic segmentation benchmarks, including Cityscapes and PASCAL VOC 2012.

Strengths:
- The idea of learning a pixel-wise embedding for encoding instance-specific visual context seems interesting, which can be used for predicting both thing and stuff categories.

- The performance of the proposed panoptic segmentation pipeline is comparable or slightly better than the published SOTA on two benchmarks.

Concerns:
- The focus of this work is a bit unclear. It seems trying to tackle two related problems, one is the learning an object instance-specific visual context representation and the other is the panoptic segmentation. However, neither of them is explored in depth or with compelling results in this paper.

- The technical contribution of this work is a bit incremental. Most of the learning framework is based on the Segment Sorting paper and its extension to instance-level supervision seems straightforward.

-  While learning embedding for object-centeric context is useful for certain applications, it also raise the question on the complexity of such holistic features and how well such a representation can generalize to different scenes.  Due to the compositionality of object and its surroundings, conceptually it seems challenging to learn such representations with limited model capacity. In particular, how well the learned representation is able to cope with an object out of its common context?

- The evaluation of the panoptic embedding is a bit lacking. First, as the method is fully supervised, all different types of object-centric context do exist in the ground-truth label maps. As such, it seems while contexts are not annotated, they should be able to learned by the model via segmentation loss? Second, the CE metric is hard to interpret as it is a KL divergence measure. It would be more clear if common retrieval performance metrics can be adopted to show its improvement. Finally, the paper only compares two representations for object-centric embedding and it would more informative to show the performance of other types of features, such as SegSort, One-stage instance segmentation, etc.

- For the panoptic segmentation, the proposed pipeline adopts a complex post-processing stage to merge the pixel representations into instance segmentation. Without detailed ablative study, it seems unclear what components contribute to the final performance. Is the learned embedding or a better merging strategy? Moreover, the common practice in panoptic segmentation literature is to conduct evaluation on both COCO and CityScapes, and report detailed metrics on PQ, SQ, RQ. Here the COCO benchmark and some metrics are missing.

Minor comments:
Sec 1: Paragraph 2: extra words at the end: "A major"

=====POST-REBUTTAL COMMENTS========

I thank the authors for the response and the efforts in the updated draft. Unfortunately,  I still think the results on the object representation is not convincing due to lack of comparisons with other embedding methods, and more needs to be done to study the generalizability of this method for complex non-street scenes.  I retain my original decision for these reasons.

---

> ### Author Response · Authors · 2020-11-24
> **We appreciate your recognition of our novelty and competitive results**
>
> We appreciate your recognition of our novelty and competitive results.  Here we respond to your insightful comments.
>
> Q1: Focus is unclear?
> * Our goal is not to simply beat the benchmarks on panoptic segmentation, but to use it to enable visual context encoding implicitly.   That is, to encode visual contexts, we need to understand objects and stuff; while developing our novel panoptic segmentation by studying the contrastive relationships between pixels and segments in terms of objects and stuff, we automatically capture the visual context implicitly.  While more work is needed to further explore visual contexts, to the best of our knowledge, there has not been any other work in panoptic segmentation that even remotely studies visual contexts.
>
> Q2: Weak technical contribution?
> * Extending SegSort from semantic segmentation to instance segmentation is highly non-trivial.  Separating a few semantic categories is much easier compared to separating two similar cars side by side, let alone recognizing thousands of cars in the dataset.  One critical difference is the need of object priors -- The model has to understand when and how segments form an object, when and how segments need to be separated into two objects.   In our current implementation, we adopt a seed assisted merging process in a coherent fashion that enables good performance, although the detailed mechanism can be replaced and simplified with a better alternative in the future.
>
> * There are two camps of approaches to panoptic segmentation.   One camp is bottom-up in the vein of agglomerative clusterings, such as DeeperLab (Yang et al., 2019) and SSAP (Gao et al., 2019).  The other is top-down in the vein of template matching, which includes all the other approaches.  The latter is the current mainstream and almost always yields much better performance.  Our work sits between these two camps.  It is the first one to bridge a bottom-up panoptic embedding process with a top-down seed diffusion process, bringing the first camp approach closer to, situationally even better than, the latter camp.
>
>
> Q3: Cope with novel contexts?
> * To answer this question, we conduct an analysis on pedestrian-centric visual context.  We first collect all the pedestrian prototypes in the Cityscapes training set. We then plot their surrounding ground truth mask at their t-SNE feature locations and the aggregated density map.  This figure is now added in the main paper as Figure 5 on page 9.  We observe interesting clusters such as pedestrians next to a car and pedestrians alone on sidewalks.
>
> * In particular, we observe the contexts are sorted automatically by the learned embedding.  By examining the aggregated density map, we can also find where the feature space is mostly empty, which points to potential novel contexts.  As a result, we notice some rare contexts on the middle left: a pedestrian is behind a clutter of a motorbike and a bike, which could possibly lead to a collision.  We believe this analysis suggests that our method is able to cope with novel contexts, which are usually isolated in the feature space.
>
>
> Q4: Contexts should be learned via annotation?  Retrieval ground truth?
> * Thanks for suggesting the retrieval metrics.  For each query object, we retrieve 20 objects with similar contexts using the ground truth mask.  We then retrieve K objects using either ours (PSS) or the baseline UPSNet.  We compute the retrieval precision@K and summarize the results in the table below.  It demonstrates that our method also outperforms UPSNet by a large margin, with a relative gain from 57% to 77%.
>
> | K | PSS | UPSNet | our relative improvements |
> | ------------- |:-------------: |:-------------: |:-------------: |
> | 1 | 26.06% | 14.70% |  +77.28% |
> | 5 | 21.77% | 13.69% | +59.02% |
> | 10 | 19.67% | 12.48% | +57.61% |
>
>
> Q5: COCO benchmarks?  SQ and RQ?
> * We have a rather limited compute.  Training on MSCOCO also takes 4 days on 4 GPUs, as mentioned by the UPSNet work (Xiong et al., 2019).  Such a long turn-around prevents us from conducting solid experiments in such a short rebuttal time frame.  However, different methods often perform similarly on both Cityscapes and MSCOCO.  The experimental results we conducted on Cityscapes and VOC in the paper could serve to validate our approach given our limited compute resources.
>
> * We omit the SQ and RQ metrics on Cityscapes, only because we find their values and trends are consistent with PQ stuff and PQ thing, respectively.  Below is the complete set of metric numbers on Cityscapes.
>
> |  Method | PQ | SQ | RQ | PQ stuff | PQ thing |
> | ------------- |:-------------: |:-------------: |:-------------: |:-------------: |:-------------: |
>   | UPSNet |  59.1% | 79.4% | 72.6% | 62.6% | 54.2% |
>   | Ours |      58.7% | 80.2% | 68.7% | 63.7% | 51.7% |

---

### Official Review · AnonReviewer2 · 2020-10-29
**This work proposes a metric-learning-based framework, panoptic segment sorting, for performing panoptic segmentation. The proposed approach achieves competitive performance on cityscapes and pascal voc.**

**Rating:** 5
**Confidence:** 4

**Review:**

Pros

This work is well written and easy to understand. The idea of assembling over-segments sorting, segments merging, dynamic partitioning, and seed selection is interesting, which can be applied to many downstream tasks, like panoptic segmentation and instance relationship modeling.

Cons

1 This work emphasizes that the proposed approach can encode and discover object-centric context. Besides, it further designs experiments to compare the context errors between UPSnet and the proposed one. However, I am not quite sure what is the use of object-centric context? Is it meaningful for some applications?

2 Even I consider the proposed idea is interesting, the overall pipeline seems more complex than many end-to-end panoptic segmentation frameworks such as UPSnet and panoptic-deeplab. Besides, the work does not achieve better performance than previous solutions.

3 How about the generalization ability of this work? For example, can PSS be trained on Cityscapes and applied to coco-stuff based on a few-shot setting (i.e. only a few training samples are available)?

---

> ### Author Response · Authors · 2020-11-24
> **Thank you for acknowledging that our proposed idea is interesting and can be applied to many downstream tasks**
>
> Thank you for acknowledging that our proposed idea is interesting and can be applied to many downstream tasks.  Now we address the concerns you raised.
>
> Q1: The use of object-centric contexts?
> * There are at least 4 possible challenging and exciting applications of object-centric contexts.
> 1) Fine-grained semantic categories by object-centric contexts.  As suggested in the paper, pedestrians can be further categorized by where they appear (e.g. in a busy intersection vs. an empty sidewalk) or what actions they are taking (e.g. window shopping vs. hailing a taxi).
> 2) Data-driven scene context organization.  Scene contexts can be hard to describe in words and yet very useful in practical applications such as autonomous driving.  Our work can be used to actively slice the training data according to scenes of interest without any scene labels.  For example, tail events and corner cases can be identified from a large unlabeled dataset and new models can be trained to improve the accuracy of predictions for those rare yet critical cases.
> 3) Contexts for object recognition.  It has been long recognized that an object can be readily identified not by what foreground it is itself, but by what background it is embedded into (Torralba, IJCV 2003).  Object-centric contexts can be useful for speeding up and improving the accuracy of object detection and recognition.
> 4) Anomaly detection.  We can analyze the statistics of object-centric contexts for a specific query object category and find rare or abnormal events.  Please refer to our answer to R4’s Q3, where we add a new t-SNE visual analysis on page 9 in the main paper to showcase how our learned embedding copes with anomaly contexts.
>
> Q2: The proposed pipeline is more complex?
> * Yes, while the conventional pixel-wise labeling pipeline appears simpler and more straightforward, our contrastive learning approach based on visual relationships is more flexible and generalizable, in terms of adapting to new domains and open-world settings.  Our approach is novel and not as familiar;  while it might seem complex with several small components, it is after all the first attack along this new direction and these small components can be simplified and replaced with more research.
>
>
> Q3: Generalizability?
> * Yes, generalizability is a major benefit of our contrastive learning approach.  However, Cityscapes and MSCOCO contain wildly different appearances and scenes, so it’s very hard to show the difference of generalizability using the two.  Usually, the ImageNet pre-trained features are still dominant.  More generalizability experiments can be explored and demonstrated with a completely unsupervised learning formulation that we will leave as future work.

---

### Decision · Program_Chairs · 2021-01-07
**Final Decision**

**Decision:**

Reject

**Comment:**

The reviewers were split (with all scores hovering around borderline) and I found it difficult to reach a conclusion. I like the paper, and agree with the authors that it may offer an interesting "middle ground" between bottom-up and top-down approaches. On the other hand, I was concerned with some of the execution flaws that were brought up in the reviews, in particular, insufficient comparisons to other embedding methods, lack of results on COCO, and to a significant degree, lack of focus in presentation. I think this could be a much stronger paper, and it will benefit from additional time to line up those missing components.
To clarify the concern re: experiments on COCO, since the authors bring up computational constraints: I agree that running these experiments during the rebuttal period is not a reasonable expectation. But the conclusion is that these results should have been in the original submission! Semantic/panoptic segmentation is now a very mature area, and COCO (along with CityScapes) is one of the standard benchmarks. (BTW, "different methods often perform similarly"  on COCO and CityScapes, but not always -- partially due to the significant differences in the statistics of the two datasets). I don't think it's reasonable to have a submission in this area which does not include results on it, since it makes it very hard to assess how much empirical progress is being made.